# Vegan and Vegetarian Soups Are Excellent Sources of Cholinesterase Inhibitors

**DOI:** 10.3390/nu16132025

**Published:** 2024-06-26

**Authors:** Dorota Gajowniczek-Ałasa, Ewa Baranowska-Wójcik, Dominik Szwajgier

**Affiliations:** Department of Biotechnology, Microbiology and Human Nutrition, University of Life Sciences in Lublin, Skromna 8, 20-704 Lublin, Poland; ewa.baranowska@up.lublin.pl (E.B.-W.); dominik.szwajgier@up.lublin.pl (D.S.)

**Keywords:** AChE inhibitors, BChE inhibitors, Alzheimer’s disease, plant-based diet, vegan, vegetarian diet

## Abstract

Background: The cholinesterase theory stands as the most popular worldwide therapy for Alzheimer’s disease (AD). Given the absence of a cure for AD, a plant-based diet has been repeatedly shown as positive in the prevention of AD, including exploring ready-made products in stores and the development of new functional foods. Goal: This study compared the anti-acetyl- and butyrylcholinesterase activity of thirty-two Polish market soups and five newly formulated soups intended to be functional. Additionally, the research aimed to assess the significance of animal content, distinguishing between vegan and vegetarian options, in cholinesterase inhibition. Materials and methods: The anticholinesterase activity was investigated using a spectrophotometric method, and the inhibitory activity was expressed as % inhibition of the enzyme. The study categorized soups into three groups based on ingredients: those containing animal-derived components, vegetarian soups and vegan soups. Results: Soups exhibited varying levels of activity against acetylcholinesterase (AChE) and butyrylcholinesterase (BChE), indicating differences in their compositions. Composition appeared to be the primary factor influencing anticholinesterase activity, as soups within each group showed significant variability in activity levels. While some commercial soups demonstrated notable anticholinesterase activity, they did not surpass the effectiveness of the optimized soups developed in the laboratory. Certain ingredients were associated with higher anticholinesterase activity, such as coconut, potato, onion, garlic, parsley and various spices and herbs. Conclusions: Vegetarian and vegan soups exhibited comparable or even superior anticholinesterase activity compared to animal-derived soups, highlighting the importance of plant-based ingredients. The study underscores the need for further research to explore the mechanisms underlying the anticholinesterase activity of soups, including the impact of ingredient combinations and processing methods.

## 1. Introduction

Recent research has illuminated the intricate nature of Alzheimer’s disease (AD), a neurodegenerative disorder marked by cognitive decline and memory impairment, influenced by various underlying factors contributing to its progression. These factors encompass cholinergic dysfunctions, amyloid beta (Aβ) toxicity, tau protein hyperphosphorylation, synaptic dysfunction, oxidative stress and neuroinflammation [1]. The very complex nature of AD’s pathological mechanisms, combined with limited knowledge, has presented significant hurdles in developing effective prevention methods for this incurable disease.

The cholinesterase theory stands as a prominent hypothesis that plays a vital role in the therapy of AD. This theory centers on the role of a neurotransmitter acetylcholine (ACh) and the enzymes responsible for its breakdown—acetylcholinesterase (AChE) and butyrylcholinesterase (BChE)—in the development and progression of AD [2]. Acetylcholine plays a crucial role in proper brain function, especially in processes related to learning, memory and cognitive function. In AD, there is a significant reduction in ACh levels due to the degeneration of cholinergic neurons. AChE and BChE rapidly degrade ACh, limiting its availability for neurotransmission [3]. The cholinesterase theory suggests that inhibiting both enzymes could potentially increase ACh levels, thereby enhancing neurotransmission and cognitive function in individuals with AD [2]. Presently, cholinesterase inhibitors are the most widely prescribed drugs for AD treatment. However, researchers are actively investigating natural sources of inhibitors in the quest for a safer and more effective alternative medication [4] along with a well-balanced diet rich in natural cholinesterase inhibitors.

Plant-based diets, such as vegetarian or vegan ones, have the potential to impact the circulating levels of inflammatory biomarkers, consequently lowering the susceptibility to chronic diseases [5]. Various lifestyle and dietary patterns have undergone extensive scrutiny to pinpoint the optimal preventive measures against AD and related dementias. Plant-based dietary patterns have consistently demonstrated a positive correlation with the prevention of and reduction in the likelihood of AD and related dementias, as shown in multiple studies [6].

Therefore, the aim of this study was to test thirty-two Polish market soups and five newly formulated soups intended to be functional.

## 2. Materials and Methods

### 2.1. Chemicals

AChE (C3389), acetylthiocholine iodide (ATCh, 01480), BChE (C7512), BTCh (B3128), 5,5′-dithiobis(2-nitrobenzoic acid) (DTNB, D8130), eserine (physostigmine, P1600000) and Tris-HCl buffer (1185-53-1) were purchased from Sigma-Aldrich (Saint Louis, MO, USA/Poznań, Poland).

### 2.2. Materials

Thirty-two commercial soups were bought in local stores in Lublin, Poland. Detailed information regarding these soups can be found in the Appendix A and the producers can be obtained from the corresponding author. Five soups were meticulously developed and produced in our laboratory, and a comprehensive description can be found in our previous publication [7].

### 2.3. Preparation of Soup Samples for Anticholinesterase Activity Testing

The tested sample (1 mL) was carefully pounded in a laboratory mortar to obtain homogeneity and completed with 1 mL of Tris-HCl buffer (100 mM/mL, pH 8.0). The dry mass of the studied samples (0.5–1 g of the sample dried in an Eppendorf tube) was determined by drying at 90 °C for at least 18 h (until it reached a constant weight; results are available by request).

### 2.4. Inhibition of AChE and BChE

Enzyme inhibition was evaluated using the method established by Ellman et al. [8], with our own modifications [9]. To assess the solutions, we mixed 0.050 mL of the sample under investigation with 0.035 cm3 of ATCh (1.5 mM/mL), 0.175 mL of 0.3 mM/mL 5,5′-dithiobis-(2-nitrobenzoic acid) (DTNB) containing 10 mM/mL NaCl and 2 mM/mL MgCl2 and 0.02 mL of either AChE or BChE solution (0.2 U/mL). The sample volume was then adjusted to 0.345 mL using Tris-HCl buffer (100 mM/mL, pH 8.0), with all reagent solutions prepared in the same buffer. We tracked the absorbance increase at 405 nm and 22 °C using a 96-well microplate reader (Tecan Sunrise, Grödig, Austria). Alongside the samples under scrutiny, we also examined negative samples, substituting the studied samples with Tris-HCl buffer. We monitored the spontaneous hydrolysis of the substrate by analyzing blank samples containing DTNB and ATCh, topped up to 0.345 mL with Tris-HCl buffer. To ensure precise outcomes, we subtracted the absorbance originating from the blank sample and the background of the tested sample from the absorbance of the test sample. A positive control was performed using eserine (at 100 nM/dm^3^, instead of the tested sample) in order to check the experimental protocol. The ability of eserine to inhibit the activity of AChE and BChE was confirmed, in comparison with a negative control containing buffer instead of the tested sample.

Each sample was analyzed in at least four replicates, and all solutions used in a set of analyses were prepared in the same buffer to maintain consistency. The data obtained from these tests were expressed as mean values (±standard error of mean (SEM)).

### 2.5. Statistical Analysis

The routine statistical tests including average values and standard deviation were tested. Following a significant one-way ANOVA test, Tukey’s HSD post-hoc test was performed to identify differences between means (significant differences identified a *p* < 0.05). All tests were performed using Statistica Software (Version 13.1, StatSoft, Cracow, Poland).

## 3. Results and Discussion

Soups in Figure 1 (anti-AChE activity) and Figure 2 (anti-BChE activity) were grouped into three categories (from left to right): soups containing ingredients of animal origin (6, 10–16, 24–26, 8), vegetarian soups (1–5, 7, 23, 27, 29) and vegan soups (8, 9, 17–22, 30–37). Soups 33–37 were created in our laboratory as a result of optimization studies and were the most efficient soups among, in total, 18 soups that were newly formulated and intended to be functional in AD. All soups were presented in our previous publication [7].

Moving on to our original results in the presented paper, taking into consideration separately each individual group of soups (i.e., “animal-derived”, vegetarian and vegan), it can be seen that the products surprisingly very much differ in anti-AChE and anti-BChE activity (Figure 1 and Figure 2). In each of the three groups of products, there are soups with dramatically low anti-AChE and/or anti-BChE activities, compared to competitors from the same group (no. 3 or 28 for anti-AChE and no. 12, 27 or 36 for anti-BChE activity). This suggests that the composition of soups is the main, if not the sole, factor that forms the anticholinesterase activity, as all soups underwent a similar production process and thermal preservation and all have a similar physical appearance (see Appendix A).

The direct comparison of soups belonging to “animal-derived”, vegetarian and vegan groups is very difficult due to the complex and differentiated composition of each soup. Moreover, we can also only assume that the composition of the soups is as declared by the manufacturer, as we do not have any tool to verify it. However, it is not the point of this work to verify the real composition of commercial soups. We aimed to compare the market soups with each other but also with optimized soups, which we presented in detail in the previous publication and among which we selected the most effective soups for the current work. In other words, we wanted to check the activity of several dozen commercial soups (all that could be obtained on the Polish market) as they are and compare this activity with our optimized soups in order to diagnose the situation we found on the food market in Poland.

Despite the huge variety of soup compositions that we analyzed in this work, it may be worth considering whether there are “universal” soup ingredients that, regardless of the type of soup and its manufacturer, contribute significantly to the activity under consideration. A search for the most active ingredients in soups can be performed based on the results of this study. Among commercial soups, the most efficient ones can also be identified. In the case of anti-AChE activity, the most efficient “animal-derived” soups were no. 6 (coconut ginger chicken soup), no. 10 (potato soup), no. 25 (pea soup) and no. 26 (red curry soup). These soups were a combination of polyphenolic and terpene compounds originating from coconut, potato, onion, including shallot, peanuts, galangal, cumin, soy components, pak choi cabbage, celeriac, oyster mushroom, mung bean, various types of pepper, red curry, coriander, kafir leave (coconut ginger chicken soup 6 or red curry soup 26); potato, celeriac, soy isolates usually containing polyphenolic fractions, dill and spices and herbs, which were not listed in detail (potato soup no. 10); and pea, potatoes, celeriac, parsley and spices, which were not listed in detail (pea soup no. 25). Highly active vegetarian soup 5 contained several very active anti-AChE components like cauliflower, potato, celery, parsley, leek and dill. Last but not least, commercial vegan soups (17 and 30) were very efficient and were rich sources of components from green peas, coconut, potato, leak, parsley, garlic, black pepper, onion, dill, lemon and mint. Although the above-mentioned commercial (“animal-derived”, vegetarian and vegan) soups were efficient as sources of AChE inhibitors, they did not reach the activity of optimized soups 35–37, which were the most efficient soups in the whole study. The soups 35–37 were selected among 18 other soups designed and produced in our laboratory, as presented in [7], our cited work; soups 33–37 were designed and produced in our laboratory in the “anticholinesterase-oriented” manner in order to use components carefully studied and selected in our laboratory after the screening tests from the past. That is, components were specifically selected in order to exert the intended activity, as well as to ensure that the sensory characteristics of these soups were acceptable. However, only soups 35–37 exceeded commercial soups and soups 33 and 34 did not stand out positively from commercial soups. In soups 33–37, we used simple components like leek, known for its phenolic constituents, antioxidant and enzyme inhibition activities [10], potatoes, onions, garlic and parsley leaves in order to form a good foundation for functional foods. It is worth noting that soups produced by us and the above-listed commercial soups contained selected common components. However, our newly designed soups 35–37 also contained some other components like green asparagus, apple, sea buckthorn fruit, cinnamon, boletus (*Imleria badia*) and blackthorn. There is a considerable number of papers published in the past, pointing out that the above-mentioned components of foods are rich sources of cholinesterase inhibitors. In vitro experiments have verified the antioxidant properties and enzyme inhibitory capabilities of blackthorn fruits [11]. Sea buckthorn berries are in harmony with the pursuit of natural components in preventing non-communicable diseases, in line with the cholinergic theory [12]. Mushrooms contain compounds that can inhibit acetylcholinesterase, potentially offering neuroprotective effects [13]. Green asparagus extract exhibited a reduction in AChE levels [14]. The ability to inhibit AChE by apples is widely known [15]. Peas are known for their ability to inhibit cholinesterase [16], as are more exotic soup ingredients like coconut, due to its phenolic compounds [17], galangal, which is a multipotent spice [18], or the typical medicinal plant in Asian cuisine, kaffir lime leaves [19]. Spices may provide health benefits and are not just a flavor in food. Curcuma, cinnamon and black pepper present anticholinesterase potential [20,21].

As for anti-BChE activity (Figure 2), selected commercial soups stand out. Soups 10, 17, 25, 35 and 37, already presented above as a rich source of AChE inhibitors, also very efficiently provided BChE inhibitors. However, other soups can also be pointed out in this context. “Animal-derived” soups 11 and 13 contained inhibitors, probably due to the presence of some components that were found in the soups discussed earlier, together with some new ones: coconut, onion, potato, chestnuts, green curry, ginger, garlic, limes, soy, coriander, kaffir lime, pea, pepper and marjoram. Interestingly, vegetarian soups were ranked, in terms of inhibition of BChE, equal to soups containing meat components. This obviously confirms the complete lack of cholinesterase inhibitors in raw materials of animal origin, which we have observed over the years in our research on this topic. In other words, the anticholinesterase activity of soups depends only on the quality of plant components and their content in the soup formulation.

The most efficient vegetarian soups were 1, 4 and 23, with putative bioactive ingredients derived from potatoes, beetroot, white cabbage, white beans, onion, dill, bay leaf, pepper, parsley root and leaves, leak, garlic and unspecified spices (soups 1 and 4) and potatoes, celeriac, dill, parsley, onion, parsnip and unspecified spices (soup 23). As for vegan soups, except soups 17, 35 and 37, whose compositions were presented above, the highest activities were exhibited by soups 9 and 34, and the putative active components of these soups can be potato, onion, celeriac, garlic, basil, green asparagus, parsley, calendula and black pepper. Some of these ingredients appeared in the most active soups with anti-AChE activity and were discussed earlier. As for others, like basil and calendula, there are some reports showing that these plants are rich sources of cholinesterase inhibitors. For example, Tamfu et al. [22] showed that basil very effectively inhibited AChE and BChE activity. Calendula has the potential to inhibit cholinesterase, which varies along with extraction methods [23]. Potato’s potential for inhibiting cholinesterase is well researched [24,25], as is onion’s [26] and parsley’s [27].

## 4. Conclusions

This work screened several dozen dinner soups for anti-AChE and anti-BChE activity. We showed that, among 37 soups (Appendix A), selected commercial soups as well as optimized soups produced in our laboratory exerted high anticholinesterase activity. Some of the soups were efficient both towards AChE and BChE, whereas selected products preferred one of the enzymes. These observations are in agreement with past studies showing differences in the degree of interactions of the food matrix (cholinesterase inhibitors) with both enzymes.

Our study utilized a simple in vitro spectrophotometric method. However, the presented studies are, in our opinion, informative and original, and the continuation is planned. It should be strongly emphasized that individual responses to dietary patterns can vary, and factors such as overall diet quality, specific food choices and lifestyle factors also play a role. Additionally, more research is needed to fully understand the mechanisms by which plant-based diets impact inflammation and chronic disease risk. Last but not least, not only the ingredients of the soup may influence the ability to inhibit cholinesterase. The combination of ingredients and the method of processing is also important. These topics will be studied in the near future.

## Figures and Tables

**Figure 1 nutrients-16-02025-f001:**
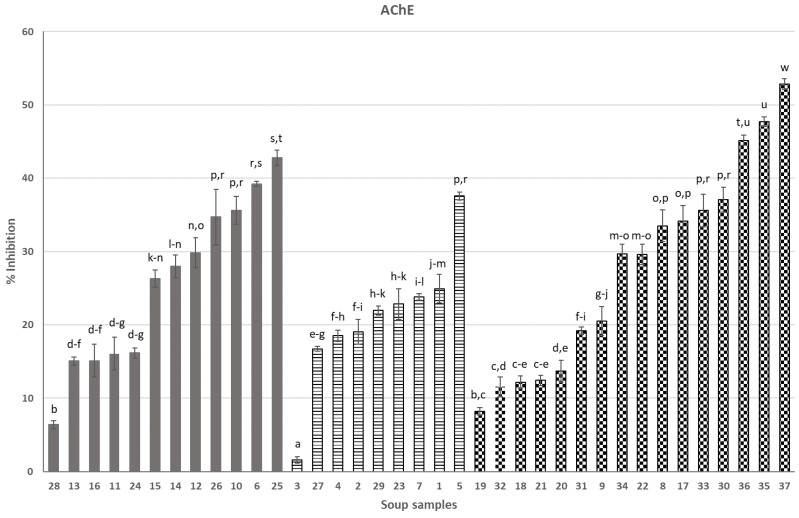
Effect of soups on AChE activity (%): 1—Ukrainian borsch, 2—pumpkin soup, 3—tomato soup, 4—Ukrainian borsch, 5—cauliflower, 6—Thai, 7—vegetable, 8—pickled cucumber, 9—tomato with basil, 10—potato, 11—Vietnamese soup, 12—Hungarian goulash, 13—pea soup with sausage, 14—tomato with chicken, 15—sour soup, 16—Thai, 17—green pea cream, 18—cream soup with Italian tomatoes, 19—orange pumpkin and mango cream soup, 20—pumpkin cream soup with cinnamon, 21—tomato cream with basil, 22—traditional pickled cucumber, 23—cucumber with potato, 24— barley soup, 25—pea with potato and bacon, 26—red curry, 27—pickled cucumber with dill, 28—sour soup with sausage, 29—broccoli, 30—cream pea with mint, 31—beetroot with coconut milk, 32—Italian pureed tomato cream soup, 33—leek soup, 34—asparagus A, 35—asparagus B, 36—sea buckthorn soup, 37—boletus soup; animal-derived—grey bars, vegetarian—stripped bars, vegan—checked bars; different letters over the bars (a, b…) denote significant differences (*p* < 0.05) between the samples.

**Figure 2 nutrients-16-02025-f002:**
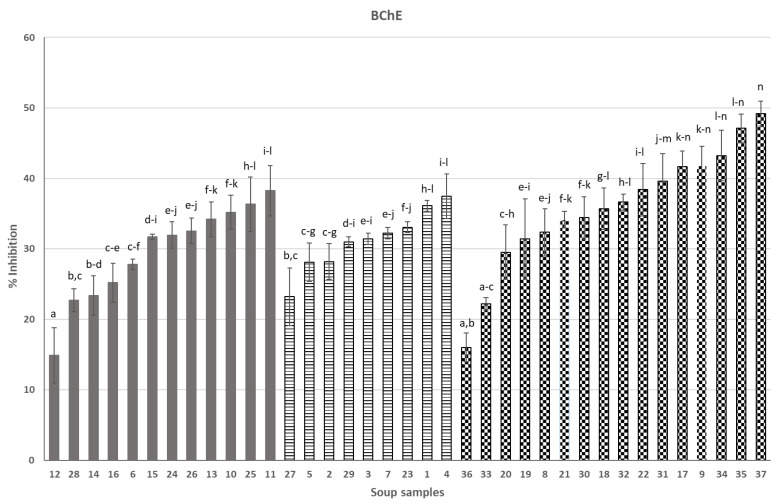
Effect of soups on BChE activity (%): 1—Ukrainian borsch, 2—pumpkin soup, 3—tomato soup, 4—Ukrainian borsch, 5—cauliflower, 6—Thai, 7—vegetable, 8—pickled cucumber, 9—tomato with basil, 10—potato, 11—Vietnamese soup, 12—Hungarian goulash, 13—pea soup with sausage, 14—tomato with chicken, 15—sour soup, 16—Thai, 17—green pea cream, 18—cream soup with Italian tomatoes, 19—orange pumpkin and mango cream soup, 20—pumpkin cream soup with cinnamon, 21—tomato cream with basil, 22—traditional pickled cucumber, 23—cucumber with potato, 24—barley soup, 25—pea with potato and bacon, 26—red curry, 27—pickled cucumber with dill, 28—sour soup with sausage, 29—broccoli, 30—cream pea with mint, 31—beetroot with coconut milk, 32—Italian pureed tomato cream soup, 33—leek soup, 34—asparagus A, 35—asparagus B, 36—sea buckthorn soup, 37—boletus soup; animal-derived—grey bars, vegetarian—stripped bars, vegan—checked bars; different letters over the bars (a, b…) denote significant differences (*p* < 0.05) between the samples.

## Data Availability

The raw data supporting the conclusions of this article will be made available by the authors on request.

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
