# Peer review of "Vegan and Vegetarian Soups Are Excellent Sources of Cholinesterase Inhibitors"

_nutrients, 2024, doi:10.3390/nu16132025_

Round 1

Reviewer 1 Report

Comments and Suggestions for Authors

The manuscript "Vegan and vegetarian soups are excellent sources of cholinesterase inhibitors" reports a study comparing the anti-acetyl- and butyrylcholinesterase activity of 32 Polish market soups and five formulated soups. The authors noted that the optimized soups developed in the laboratory show more anticholinesterase activity effectiveness.  The ingredients such as coconut, potato, onion, garlic, parsley, and various spices and herbs are associated with the anticholinesterase activity.

The reported results can be used in further studies. 

There are a few points that can improve the visibility of the manuscript.

1 - "soups and 5 newly formulated"

"were grouped into 3 categories"

Please for numbers < 10, it is suitable to write the numbers in full. Please replace 5 with "five", "3" with "three" and so on. Please revise the whole manuscript. 

2 - Please improve Figure 2 and Figure 3. I suggest using colors to group the soups into three categories. 

3 - I strongly suggest a table with the ID of the soups and the main ingredients, perhaps in percentage.  The discussion could cite this table and add to improve the conclusion section.

Author Response

Dear Reviewer 1,

Thank you very much for your valuable comments. I have carefully followed the instructions provided in points 1 and 2. Point 1, in particular, provided me with new knowledge, as I was previously unaware of the rule about writing out numbers as "one" and "two" instead of "1" and "2." I am very grateful for this insight.

However, I am seeking further clarification on point 3. The table containing the soup IDs is included in the Supplementary file due to its large size. Would you recommend incorporating this table into the main text, or is it acceptable to keep it in the Supplementary file?

1 - "soups and 5 newly formulated" "were grouped into 3 categories"

Please for numbers < 10, it is suitable to write the numbers in full. Please replace 5 with "five", "3" with "three" and so on. Please revise the whole manuscript. 

2 - Please improve Figure 2 and Figure 3. I suggest using colors to group the soups into three categories. 

3 - I strongly suggest a table with the ID of the soups and the main ingredients, perhaps in percentage.  The discussion could cite this table and add to improve the conclusion section

Reviewer 2 Report

Comments and Suggestions for Authors

This manuscript is a comparison of the anticholesterase inhibition of different commercial and custom-made soup products in vitro. Overall, it is an interesting manuscript. I recommend the following revisions:

1. I recommend revising Figure 1 and 2 for more clarity. I am unable to understand how to interpret the letters above each bar indicating significance. More clarity in how to interpret that information is needed in the figure legend or an alternative way of presenting the data is needed. I also recommend revising the figures such that the bars for each soup type (i.e., vegetarian, vegan, animal) are different (i.e., stripes, solid, dots, etc) for ease of comprehension.

2. The figure legends for Figure 1 appears to be incorrect and should be revised. It indicated BChE activity.

3. There is no reference 1 in the bibliography.

4. In line 87, it states "detailed results on email." I am unsure if this is meant to state detailed results available by request or if something else is intended with this language.

5. It is unclear, was a positive control used in addition to a negative control for the inhibition studies. If not, it is recommended to repeat the experiment with a positive control. You can then provide stats on the comparison of inhibition compared to both positive and negative control. 

6. I recommend a table or other means of displaying the compounds common across all of the most inhibitory soups (i.e., garlic, etc.). It would help with visualizing and narrowing down items for further investigation. One could also include the reference to prior studies with the dose administered and mechanisms to look for commonalities in the table.

Comments on the Quality of English Language

Overall, the language quality is sufficient for comprehension but grammatical revision is needed. 

Author Response

Dear Reviewer 2,

Thank you very much for your valuable comments. I have carefully followed provided instructions.

  1. I recommend revising Figure 1 and 2 for more clarity. I am unable to understand how to interpret the letters above each bar indicating significance. More clarity in how to interpret that information is needed in the figure legend or an alternative way of presenting the data is needed. I also recommend revising the figures such that the bars for each soup type (i.e., vegetarian, vegan, animal) are different (i.e., stripes, solid, dots, etc.) for ease of comprehension. 

Response:

According to the instructions, I have added distinguishing marks on the bars, which has significantly improved their legibility.

Significant differences (at p<0.05) are indicated using "a" for the lowest sample value and "w" for the highest. When different letters are present above bars (e.g., "d" above one bar and "g" above another), it signifies a significant difference between those samples. Conversely, if the same letter (e.g., "h") appears above two bars, it indicates that there is no significant (at p<0.05) difference between those samples.

  1. The figure legends for Figure 1 appears to be incorrect and should be revised. It indicated BChE activity.
  2. There is no reference 1 in the bibliography
  3. In line 87, it states "detailed results on email." I am unsure if this is meant to state detailed results available by request or if something else is intended with this language.

Response to point 2,3,4:

Obviously, you were right. The changes I made are highlighted in yellow in the manuscript.

  1. It is unclear, was a positive control used in addition to a negative control for the inhibition studies. If not, it is recommended to repeat the experiment with a positive control. You can then provide stats on the comparison of inhibition compared to both positive and negative control. 

Response: a positive control was performed, using eserine (at 100 nM/dm3, instead of the tested sample), in order to check the experimental protocol. The ability of eserine to inhibit the activity of AChE and BChE was confirmed, in comparison with a negative control, which contained buffer instead of the tested sample, so the full analysis was performed.

  1. I recommend a table or other means of displaying the compounds common across all of the most inhibitory soups (i.e., garlic, etc.). It would help with visualizing and narrowing down items for further investigation. One could also include the reference to prior studies with the dose administered and mechanisms to look for commonalities in the table.

Response:  While we understand the value of identifying common compounds across the most inhibitory soups, our approach was different. Instead of analyzing individual inhibitors, we focused on creating soups by combining the best cholinesterase inhibitors identified through our analyses as well as after the literature research. A detailed description of this process can be found in our previous publication.

Our primary goal was to compare the soups based on their overall ability to inhibit cholinesterases rather than examining the mechanisms of individual ingredients. This approach has allowed us to gain valuable insights, which we are currently expanding upon in ongoing tests. Once these tests are completed, we plan to use a comprehensive table, as you suggested, incorporating various biochemical analyses.